# The Efficacy and Safety of Rituximab in ANCA-Associated Vasculitis: A Systematic Review

**DOI:** 10.3390/biology11121767

**Published:** 2022-12-06

**Authors:** Mohammad Amin Habibi, Samira Alesaeidi, Mohadeseh Zahedi, Samin Hakimi Rahmani, Seyed Mohammad Piri, Soheil Tavakolpour

**Affiliations:** 1Clinical Research Development Center, Qom University of Medical Sciences, Qom 3719964797, Iran; 2Sina Trauma and Surgery Research Center, Tehran University of Medical Sciences, Tehran P.O. Box 982166757001, Iran; 3Rheumatology Research Center, Tehran University of Medical Sciences, Tehran P.O. Box 982188220065, Iran; 4Dana-Farber Cancer Institute, Harvard Medical School, Boston, MA 02215, USA

**Keywords:** rituximab, immunopharmacology, wegner, MPO-associated vasculitis, PR3-associated vasculitis, small vessel vasculitis

## Abstract

**Simple Summary:**

ANCA-associated vasculitis (AAV) is a rare disease and manifests in different organs. The exact mechanism of AAV is not fully understood, but it is speculated that it is induced by autoantibody production against proteinase-3 and myeloperoxidase. Patients diagnosed with ANCA-associated vasculitis are challenging to manage and experience several relapses during and after treatment. This systematic review represents the efficacy and safety profile of rituximab in AAV patients and provides evidence that rituximab can be considered the treatment of choice for AAV patients, especially for challenging patients with refractory releases.

**Abstract:**

**Background and aim**: Antineutrophil cytoplasmic antibody (ANCA)-associated vasculitis (AAV) is a rare multisystem autoimmune disease developed by autoantibody production against human neutrophilic granulocytes, including proteinase-3 (PR3) and myeloperoxidase (MPO). The management of AAV patients is difficult due to the multiorgan involvement, high rate of relapse, and complications of immunosuppressive agents that make it challenging. This study aims to investigate the efficacy and safety of rituximab (RTX) therapy in patients with granulomatosis with polyangiitis (GPA) or microscopic polyangiitis (MPA) subtypes. **Method**: The PubMed/Medline database was searched for any studies related to RTX therapy in ANCA-associated vasculitis (GPA and MPA subtypes), from inception to 1 August 2022, and proceeded in accordance with Preferred Reporting Items for Systematic Reviews and Meta-Analyses (PRISMA). **Results**: Our search resulted in 1082 initial records. After the elimination of review papers, irrelevant studies, and non-English records, 223 articles were included, and the data related to the efficacy and safety of RTX therapy were extracted. Several randomized and non-randomized studies showed that RTX is an effective treatment option for patients with AAV. Most of the studies showed the very effective effect of RTX in controlling disease in AAV patients, including pediatrics, adults, and elderlies, although RTX cannot completely prevent relapse. However, maintenance therapy helps delay the disease’s relapse and causes sustained remission. Not only the licensed dose (375 mg/m^2^ intravenous per week for 4 weeks) could induce disease remission, but studies also showed that a single infusion of RTX could be effective. Although RTX could resolve many rare manifestations in AAV patients, there are few reports showing treatment failure. Additionally, few sudies have reported the unexpeted worsening of the disease after RTX administration. Generally, RTX is relatively safe compared to conventional therapies, but some serious adverse effects, mainly infections, cytopenia, hypogammaglobinemia, malignancy, and hypersensitivity have been reported. **Conclusions**: RTX is an effective and relatively safe therapeutic option for AAV. Studies on the evaluation of the safety profiles of RTX and the prevention of severe RTX-related side effects in AAV patients are required.

## 1. Introduction

Antineutrophil cytoplasmic antibody (ANCA)-associated vasculitis (AAV) is a group of autoimmune diseases characterized by the inflammation of small- to medium-sized blood vessels. Three major subgroups are granulomatosis with polyangiitis (GPA), known as Wegener’s granulomatosis), microscopic polyangiitis (MPA), and eosinophilic GPA (EGPA; also known as Churg–Strauss syndrome), which is characterized as a lack of tolerance to the main proteins of neutrophil granule, including proteinase 3 (PR3) or myeloperoxidase (MPO) [1]. It is estimated that 46–184 people in Europe are diagnosed with AAV annually, with incidence rates ranging from 2.1 to 14.4 for GPA, 2.4–10.1 for MPA, and 0.5–3.7 for EGPA, correspondingly [2]. People over the age of 60 and males are more likely to be diagnosed with AAV [3]. Five-year survival rates for GPA, MPA, and EGPA were estimated between 74% and 91%, 45% to 76%, and 60% to 97%, consecutively [4].

The necrotizing inflammation of small- to medium-sized blood vessels is the hallmark of AAV, leading to notable diverse clinical presentations [5]. Serum PR3 (PR3-ANCA) or MPO (MPO-ANCA), often correlated with the primary syndromic AAV presentations, may be used clinically to demonstrate autoimmunity. It is typical for GPA and MPA to impact capillaries in the upper and lower respiratory tracts and the renal tract. In most cases, AAV patients face life-threatening or organ-threatening illnesses; however, less severe presentations are possible. PR3-ANCA is the most common type of ANCA associated with GPA, and the disease’s clinical features include sinonasal disease, lower respiratory tract involvement, and glomerulonephritis. MPA is often linked to MPO-ANCA, and its symptoms include renal impairment and some of GPA’s manifestations but without the granulomatous inflammation that characterizes GPA [1]. The typical renal glomerular involvement of GPA patients is histologically similar to that observed in MPA.

Moreover, Wegener’s syndrome affects the other organs, including respiratory disease, and skin, eye, and joint involvement are different parts of the clinical presentation of Wegner and are included in the diagnoses. Despite microscopic polyangiitis, Wegener’s syndrome lesions are typically granulomatous. Some studies have proposed that MPO-AAV and PR3-AAV are disparate due to the different epidemiology, prognosis, and clinical features, and patients with PR3-AAV had a greater risk of relapse than MPO-AAV patients [6]. Patients in the MPO-AAV group were older and less likely than those in the PR3-AAV group to have ear, nose, and throat (ENT) or ophthalmic involvement. Moreover, blood creatinine and proteinuria were more prominent in MPO-AAV patients. The GPA and MPA degenerative alterations in the glomeruli and tubulointerstitia were indistinguishable. Patients with MPO-AAV had considerably more risk of end-stage renal disease than those with PR3-AAV [7].

The regular management of AAV includes an induction phase of 3–6 months to rapidly decrease inflammation and avoid persistent organ damage, followed by a prolonged maintenance phase after the induction of remission. After two years, treatment is discontinued in patients with sustained remission [8]. Although existing induction regimens are efficient in inducing remission in 70–90% of patients, they are accompanied by high levels of side effects, particularly in older patients and those with lung damage [9,10]. Future maintenance treatment is utilized to minimize relapses; nevertheless, relapses can develop in 50% of patients within five years from their first remission [11]. The standard treatment for newly diagnosed AAV patients comprises high-dose glucocorticoids (GC) with either cyclophosphamide (CYC) or rituximab [8]. Methotrexate (MTX) and mycophenolate mofetil (MMF) are additional therapies for people with non-threatening conditions. Although MTX and MMF successfully induce remission, they are more associated with a risk of recurrence than CYC or rituximab (RTX) treatment [12].

RTX is a monoclonal antibody that depletes CD20+ B cells, which can last for even more than 24 weeks. After FDA approval in 2011, RTX opened up a new avenue to treat AAV patients It was licensed at a dosage of 375 mg/m^2^/week for four weeks plus glucocorticoids [13]. Due to the increasing trend of RTX therapy, considered an alternative to CYC, here, in this review, we sought to investigate the aspects of RTX therapy in AAV patients, mainly GPA and MPA, to elucidate the beneficial and adverse effects of RTX and provide a clinical direction for the management of AAV patients.

## 2. Method

This systematic review was prepared in accordance with Preferred Reporting Items for Systematic Reviews and Meta-Analyses (PRISMA) [14]. EGPA was excluded from the present study since the EGPA Consensus Task Force recommendation and most clinical trials noted that EGPA is a distinct entity and should be diagnosed and managed separately [15]. Therefore, we limited the AAV patients to patients with MPA and GPA. Due to the limitation regarding the number of searched databases and the number of included studies, as well as not conducting a quality assessment process of included studies, the registration code of PROSPERO was not achievable.

### 2.1. Search Strategy

The PubMed/Medline database was searched for any studies related to RTX therapy in ANCA-associated vasculitis from inception to 1 August 2022, using the keywords “rituximab”, “AAV”, “GPA”, “anti-CD20”, and “MPA with appropriate Boolean operators (AND/OR/NOT), searching in the titles, abstracts, and Medical Subject Headings (MeSH). The search strategy was ((“Anti-Neutrophil Cytoplasmic Antibody-Associated Vasculitis” (Mesh)) OR (“Granulomatosis with Polyangiitis” (Mesh)) OR (“Microscopic Polyangiitis” (Mesh)) OR (granulomatosis Wegener (Title/Abstract)) OR (Wegener’s granulomatosis (Title/Abstract)) OR (granulomatosis with polyangiitis (Title/Abstract)) OR (ANCA associated vasculitis (Title/Abstract)) OR (microscopic poly angiitis (Title/Abstract)) OR (PR3-AAV (Title/Abstract)) OR (MPO-AAV (Title/Abstract))) AND ((rituximab (Title/Abstract)) OR (anti-CD20 (Title/Abstract))). Since the major database for medical papers is PubMed, and there are also limitations in including papers (number of references), we only searched in the PubMed/Medline database. The result of the systematic search was completed by manually adding potential articles by checking the references of references and the results of Google Scholar, using “rituximab”, “ANCA-associated vasculitis”, “granulomatosis with polyangiitis”, and “microscopic polyangiitis”. Among the studies, only those with novel data have been selected. Regarding those repetitive findings, only those with more power have been considered for discussion. Among the case reports, we only discussed those studies with unique findings.

### 2.2. Inclusion and Exclusion Criteria

All of the original articles that addressed the safety and efficacy of rituximab in AAV were included in the present systematic review. The exclusion criteria were:Studies on the eosinophilic granulomatous with polyangiitis (EGPA);Non-English studies;Any articles without clinical observations, such as review articles, conference abstracts, letters to the editor, and short communications.

### 2.3. Study Selection Process

The results of the systematic search were imported into the EndNote 20 software. Two reviewers (MA. H, and S. T) independently reviewed the retrieved studies and selected the eligible articles in a two-step process by screening the title and abstract. The full texts of screened studies were reviewed in the full-text assessment step by the same two reviewers and the final articles were included. The disagreements in the title/abstract and full-text assessment were resolved by discussion between the two reviewers.

### 2.4. Data Extraction

The full texts of included studies were investigated by two reviewers (M. Z and S. HR) and the data were inserted into a predesigned Microsoft Word^©^ worksheet. The data included the first author, year of publication, country, type of study, number of patients, sex of patients, BVAS score before rituximab therapy (BVAS stands for “Birmingham Vasculitis Activity Score”, a scoring system to assess disease activity in patients with systemic vasculitis), dose of rituximab, clinical manifestation of AAV, induction or remission regimen of rituximab, and type of remission (complete remission, partial remission, not remission). The investigated data were rechecked by a third reviewer (MA. H) and overlooked data were added to the worksheet.

### 2.5. Data Synthesis

A narrative synthesis of extracted data without meta-analysis was carried out by two reviewers (MA. H and SM. P) and summarized in the tables. Statistical analysis was not conducted due to the diversity of the characteristics of studies and incomplete reports of effect size. Therefore, the effects of the studies were estimated by using the synthesis without meta-analysis (SWiM) in systematic reviews guideline designed by Campbell et al. [16] to facilitate the reporting of the synthesized data.

## 3. Results

### 3.1. Search Results and Studies’ Characteristics

From 1082 initial records, after excluding studies with irrelevant titles (n = 747), 336 records were considered for full-text assessment. Review articles (n = 29), book chapters (n = 6), studies on eosinophilic granulomatous with polyangiitis (n = 70), overlap syndrome with other autoimmune diseases (n = 5), and those without full texts available (n = 3) were excluded from the study. Thereafter, extracted data from 223 included studies were categorized into two main categories, efficacy and safety. The selection process, in accordance with the PRISMA guidelines, is shown in Figure 1.

In total, 10 randomized clinical trials (RCT), 61 non-randomized cohorts, and 85 case reports were included. All of the RCTs registered in the US or Europe registry of clinical trials are summarized in Table 1. A summary of the non-randomized studies regarding the safety and efficacy of RTX therapy in AAV patients is also shown in Table 2. Case reports with similar findings without additional data are not used for discussion. Meanwhile, reports of the unsuccessful treatment of AAV patients with RTX and resolving rare manifestations of AAV patients by RTX are debated separately to highlight the points of RTX therapy. The included studies were discussed according to the subgroups of the regimens of RTX therapy, age specificity, and organ involvement. Findings addressing the adverse events of RTX therapy in AAV patients are summarized in Table 3, and the description of such complications is brought into context.

### 3.2. Efficacy Reports Based on Randomized and Non-Randomized Studies

Generally, there is consensus regarding the high efficacy of RTX for the treatment of AAV patients. However, studies have shed light on several critical points that bear discussing. Following the promising result of induction RTX therapy, it is increasingly being used as a maintenance option for AAV patients. In this section, the most important findings from randomized studies, non-randomized studies, and case reports are discussed.

#### 3.2.1. Randomized Studies

The randomized clinical trials registered in the U.S database (ClinicalTrials.gov, accessed on 24 October 2022) and European clinical trial registry (https://www.clinicaltrialsregister.eu, accessed on 24 October 2022) were established to investigate the efficacy and safety profile of RTX in AAV patients (summarized in Table 1). The first randomized clinical trial (RCT) study was designed to evaluate the efficacy and safety of RTX in comparison to intravenous CYC.

The RITUXVAS trial (ISRCTN 28528813) was conducted on 44 AAV patients with renal involvement comparing two regimens, including combined 375 mg/m^2^ RTX, 2 × 15 mg/kg CYC, and GCs without maintenance therapy, and 3 to 6 months’ administration of CYC followed by azathioprine (AZA). The RITUXVAS trial demonstrated that RTX was not superior to CYC for severe AAVs; both were associated with a sustained remission rate and no significant difference in early severe adverse events during 12 months of follow-up [17]. Another study (ISRCTN28528813) was conducted on RITUXVAS trial patients to investigate the long-term outcome of RTX therapy in AAV patients with renal disease. Within 24 months, the rates of the compound outcome of death, end-stage renal disease (ESRD), and relapse were not significantly different between RTX and CYC [18]. The RAVE trial (NCT00104299) also reported the non-inferiority of four infusions of 375 mg/m^2^/weekly RTX to daily CYC (2 g/kg) for the induction of remission in severe ANCA-associated vasculitis [19]. Such indifference was also confirmed by a subgroup analysis of a RAVE trial regarding 102 of 197 AAV patients with renal involvement [20]. Over 18 months, four doses of 375 mg/m^2^/w RTX with GCs were compared to 18-month consecutive therapy with 2 mg/kg/d CYC, which was replaced by 2 mg/kg/d AZA (3–6 months after CYC usage) plus GCs, and were not significantly different in renal function outcomes [20]. Specks et al. [21] also showed that four weekly doses of 375 mg/m^2^/w RTX have similar efficacy compared to the regimen including CYC (2 g/d) for 3–6 months and then azathioprine (2 g/d) for a total of 18 months. Therefore, not only is RTX not inferior to CYC in terms of the induction of remission, but patients might also benefit from long-term treatment with RTX due to decreasing the need for maintenance immunosuppression and also reduced cumulative exposure to CYC.

Some other RCTs were designed to compare the efficacy and safety of RTX to other immunosuppressive agents. The RITAZAREM (NCT01697267) is another clinical trial that was conducted on active AAV patients older than 15 years who received RTX and GCs for the induction of remission. It was shown that RTX was superior to oral AZA in terms of relapse prevention in patients who experienced relapse during at least 36 months of follow-up [22]. The SCOUT trial (NCT02169219) was a pilot clinical trial on 20 patients to investigate the efficacy of combining the RTX and an 8-week GCs regime in AAV patients to induce and maintain remission. Newly diagnosed MPA or GPA patients or relapsed patients with BVAS > 3, aged 18 to 85 years old, were recruited. As a result, short-course prednisolone combined with RTX achieved complete remission (CR) within 6 months similar to the RAVE trial with a lesser adverse event; however, the relapse rate was greater than in the RAVE trial [23].

The triple MAINRITSAN trials were conducted in France to investigate the efficacy of different regimens in the remission of AAV patients. The MAINRITSAN trial (NCT00748644) was conducted on 115 newly diagnosed or relapsed AAV patients, who were treated with either AZA (2 mg/kg/d) or 500 mg RTX (on days 0 and 14, and then on months 6, 12, and 18) as maintenance treatment following induction of remission by CYC-GCs. Serious relapse occurred in 20% of patients in the AZA group and 5% of the RTX group during 28 months of follow-up [24]. Moreover, a health assessment of AAV patients in MAINRITSAN trials showed a better quality of life in the RTX group compared to the AZA group [25]. Following 60 months follow-up of MAINRITSAN trial patients, it was shown that the rate of prolonged remission in the RTX group was superior to the AZA group (71.9% vs. 49.4%, respectively) [26].

**Table 1 biology-11-01767-t001:** Clinical Trials of rituximab therapy in AAV patients.

No.	RCT Name	Country	Therapeutic Assessment	RTX Dosage	Number of Patients	Age	Sex	Outcome	Status	Ref.
1	RITUXVAS	UK	RTX, CYC	375 mg/BSA	44	67.75	23M/21F	RTX is not superior to CYC	Completed	[17]
2	RAVE	US, The Netherlands	RTX, CYC	375 mg/BSA	197	52.75	98F/99M	RTX is not inferior to CYC	Completed	[19]
3	RITUXVAS *	UK	RTX plus CYC and GCs, CYC plus AZA	375 mg/BSA	44	NG	NG	No significant difference between RTX plus CYC and CYC/AZA	Completed	[18]
4	RAVE *	US	RTX/GC, CYC/AZA	375 mg/BSA	102	55	48F/54M	A single course of RTX is similar to conventional therapies	Completed	[20]
5	RITAZAREM	US, Europe, Australia, Mexico	RTX, AZA	1 g	170	57.8 ± 14.5	86F/84M	RTX is superior to AZA in the prevention of relapse	Completed	[22]
6	SCOUT	US	RTX, GCs	375 mg/BSA/w	20	61.5	7F/13M	Combined RTX and GCs resulted in the same CR as the RAVE trial	Completed	[23]
7	MAINRITSAN	France	RTX, AZA	500 mg	115	55 ± 13	50F/65M	More serious relapse in AZA than RTX	Completed	[24]
8	MAINRITSAN2	France	RTX, RTX	500 mg	162	60.5	68F/94M	No difference between adjusted infusion and organized infusion of RTX	Completed	[27]
9	MAINRITSAN3	France	RTX, Placebo	500 mg biannual	97	63.9	34F/63M	Lower rate of relapse in RTX compared to placebo	Completed	[28]
10	MAINTANCAVAS	US	RTX	2 × 1 g based on B cell repopulation	200	-	-	-	Recruiting	-
11	LoVAS	Japan	RTX+ low dose GC, RTX+ high dose GCs	375 mg/BSA	140	73.5	80F/54M	No significant difference between high-dose and low-dose GCs plus RTX regimens	Active	[29]
12	ENDURRANCE-1	The Netherlands	RTX plus CYC, RTX	2 × 1 g	47	-	-	No results available	Active, recruiting	-
13	RITUXGOPRO	France	RTX, Placebo	NG	106	-	-	-	Recruiting	-
14	COMBIVAS	UK	Belimumab, RTX	2 × 1 g	31	-	-	No results available	Active, not recruiting	-
15	SATELITE	France	RTX, Tocilizumab, Abatacept	375 mg/BSA, 500 mg	42	-	-	No results available	Not yet recruiting	-
16	RITUXGOPRO	France	RTX, placebo	2 × 1 g	106	-	-	No results available	Recruiting	-

MAINTANCAVAS: Maintenance of ANCA Vasculitis Remission by Intermittent Rituximab Dosing; LOVAS: Low-Dose Glucocorticoid Vasculitis Induction Study; SATELITE: Salvage Therapy to Treat Patients with Granulomatosis with Polyangiitis; RITUXGOPRO: Rituximab in the Treatment of Good Prognosis Microscopic Polyangiitis; NG: not given. The asterisk (*) on RAVE (RAVE) is referred to another clinical trial that conducted on AAV patients of RAVE trial with renal involvement. The RITUXVAS is the same RITUXIVAS clinical trial with 2-year results of a trial.

A phase 3 clinical trial was conducted to assess and compare the efficacy of the conventional method of the infusion of RTX and the specific infusion of RTX. The MAINRITSAN2 trial (NCT01731561) was conducted on 162 adult AAV patients in complete remission. All the patients received 500 mg RTX, which for patients in the conventional group was administrated on days 0 and 14 after randomization, and then on months 6, 12, and 18. The second group received RTX at randomization, which was re-administered whenever B-cells or ANCA titer were significantly increased. The MAINRITSAN2 trial demonstrated no significant difference between the efficacy of the customized infusion and the specified infusion of RTX in relapse rate [27]. The MAINRITSAN3 trial (NCT02433522) was carried out on 97 AAV patients in the CR of MAINRITSAN2 trial patients that experienced CR with no major relapse, with a median age of 64.6 ± 10.7 years old. The patients were randomized to the placebo or RTX group for maintenance treatment. The MAINRITSAN3 trial was conducted to increase the follow-up duration of patients under the MINRITSAN2 trial to obtain the optimal time of maintenance therapy with RTX. In this study, four doses of 500 mg RTX within 18 months (every 6 months) were infused and compared with a placebo with 28 months of follow-up. The study showed that compared to the placebo group, prolonged treatment with RTX resulted in a lower rate of disease relapse [28]. The LoVAS clinical trial (NCT02198248) was designed to evaluate the effectiveness of prednisolone administration in 140 RTX-treated (lymphoma protocol) AAV patients; 0.5 mg/kg/day and 1 mg/kg/day of prednisolone were considered as low-dose and high-dose regimens, respectively. Within 6 months of follow-up, 71% and 69.2% of patients in low-dose and high-dose GCs groups experienced remission, respectively, showing no significant differences between the two regimens in the induction of remission [29]. The other ongoing clinical trials registered in the US or European clinical trial registry are summarized in Table 1.

#### 3.2.2. Non-Randomized Studies

Non-randomized studies mainly focused on the efficacy of RTX in patients with different clinical features. As one of the first studies, Keogh et al. [30] reported nine refractory GPA patients, who were successfully treated with RTX with the lymphoma protocol. Both B-lymphocyte depletion and complete clinical remission were reported after three months. The successful treatment of eight GPA and two MPO patients with RTX with the lymphoma protocol lasting for a median of 33.5 months was also reported. All patients experienced remission within 6 months; however, relapse occurred in three patients and was resolved with repeated RTX infusion, resulting in a new sustained remission [31]. A clinical study on seven refractory AAV patients with a median age of 57 years old was conducted for the long-term assessment of the safety and efficacy of RTX. It was shown that RTX provided CR in six patents and the BVAS score was significantly decreased following 12 months of 4 × 375 mg/m^2^/w [32]. The BIOGEAS group in Spain studied 196 patients with systemic autoimmune diseases, including 19 cases of AAV. They showed that RTX therapy was associated with remission in 77% of severe refractory patients and response was more prominent in Sjogren, myopathies, systemic lupus erythematosus (SLE), and cryoglobulinemia [33]. Several studies provide evidence that RTX is considered an induction of remission agent for refractory vasculitis [30,34,35,36,37,38], severe refractory GPA [39], non-responders to conventional therapies [31], and the re-treatment of relapsed AAV [40], or as maintenance therapy to provide long-term remission [39,41,42,43,44]; however, RTX cannot completely prevent relapse [45].

#### 3.2.3. Case Reports

The first report of using rituximab for a GPA patient dates back to 21 years ago and resulted in the extension of studies [46]. A non-responder to conventional treatments was successfully and safely treated with RTX with the lymphoma protocol. The second patient was reported four years later and showed effective and safe remission induction [47]. In 2008, a Wegner patient resistant to conventional therapies was reported, who achieved partial remission following four doses of 500 mg RTX, which hampered pulmonary worsening [48]. Years later, the successful treatment of more complicated GPA patients, such as refractory GPA associated with several different clinical manifestations, including scleritis [49,50,51,52,53], orbital involvement [54,55,56], intestinal perforation [57], massive intracerebral hemorrhage [58], meningeal involvement [59,60], peripheral ulcerative keratitis (PUK) [61,62], and subglottic stenosis [63] were reported. Several case series reported GPA patients treated with RTX based on the lymphoma protocol [64,65,66,67,68,69]; however, one study showed that RTX is not associated with clinical and serological refractory GPA improvement [70]. Some studies have suggested that RTX can be considered an effective replacement therapy in severe GPA patients for whom conventional therapies are contraindicated [67] or who have a poor response to conventional therapies [71]. RTX also was found to be able to induce long-term remission, up to 3 years of disease remission in a refractory GPA patient in one study [72], and in another study, up to 38 and 48 months after RTX administration [73]. Some other interesting cases also suggested that RTX could be effectively used to treat refractory childhood onset [74,75], renal transplantation [76], and even ANCA-negative GPA patients [77,78,79].

### 3.3. Pediatrics and Elderly

Although the clinical manifestation of AAV patients in elderly and young people is similar, the severity of AAV seems to be different based on age, whereas elderly patients are shown to be at higher risk of mortality and morbidity due to AAV [80]. Adults comprise a higher proportion of AAV patients than pediatrics [81]. Thietart et al. [82] evaluated RTX for induction and maintenance therapy in AAV patients older than 75 years. It was shown that RTX induction therapy resulted in remission in 86.4% of patients and the relapse rate was lower when RTX was used as a maintenance regimen compared to induction therapy. Another study on 31 AAV patients aged older than 60 demonstrated that RTX can induce remission in 96.7% of patients, but is associated with a high rate of infection [83].

Regarding pediatrics, PePRS (NCT01750697) is a non-randomized, uncontrolled phase II clinical trial of rituximab therapy which was carried out in severe AAV pediatric patients. This study showed that an infusion of 375 mg/BSA RTX on days 1, 8, 15, and 22 resulted in clinical improvement, which was effective for the induction of remission or as maintenance therapy [84]. Another phase II global clinical trial was designed to evaluate the safety and efficacy of RTX therapy in pediatrics with GPA and MPA. The patients received 4 × 375 mg/BSA/weekly and it was shown that RTX is effective, with a similar overall safety profile to adults, which can be attributed to the adjustment of RTX dosage to body surface area [84]. Combined treatment with tapered GCs and CYC followed by 4 × 375 mg/BSA RTX as induction therapy in nine AAV patients with glomerulonephritis (GN) was effective and all patients experienced remission [85].

### 3.4. Dose

A multicenter study in the UK compared two different RTX dosages for induction therapy, 4 × 375 mg/m^2^ and 2 × 1 g, with a two-week interval, on 65 refractory AAV patients. In this survey, no difference between the two regimens was observed and response to RTX was achieved in 98 percent of patients [86]. Moreover, another study investigated low-dose RTX therapy as an induction regimen and showed that low-dose RTX therapy was effective in refractory AAV patients, which has a beneficial effect because of the expensive price [87]. Satio et al. [88] investigated a single dose of 375 mg/m^2^ and confirmed that a single dose of RTX is effective in MPA and was associated with decreasing economical damage. However, Moog et al. [89] showed that a single dose of RTX combined with other immunosuppressive therapies is not as effective as a licensed RTX therapy regimen. Yusof et al. [90] evaluated 35 relapsing AAV patients treated with 2 × 1 g RTX and showed that RTX therapy was associated with long-lasting remission. Moreover, no difference between the infusion amount of RTX in the two main RTX therapy regimens (4 × 375 mg/m^2^ and 2 × 1 g) was observed in terms of response to treatment and relapse [91].

### 3.5. Comparing RTX to Conventional

Some studies have investigated and compared the efficacy of RTX and other therapeutic options. RTX is another option for patients with contraindications for conventional therapies. For example, the results of a study on 11 refractory patients with PR3-AAV or for whom CYC was inadvisable showed that RTX (4 × 375 mg/m^2^) was able to induce remission [92]. Moreover, Roccatello et al. [93] showed that six doses of 375 mg/m^2^ RTX were effective in long-term remission without maintenance therapy and can be considered for patients who are intolerant to conventional immunosuppressive therapies. A retrospective study on 92 patients with GPA has shown that both RTX and CYC significantly reduced the BVAS score and successfully induced remission. Moreover, no significant difference was observed between RTX and CYC in terms of adverse events and death; however, the relative rate of complications was higher in the CYC group [94]. A retrospective study on 153 AAV patients with GN showed no significant difference between RTX and CYC for the induction of remission [95]. Tavakoli et al. [94] also reported the similar efficacy of RTX and CYC in the induction of remission.

It has been accepted that RTX could be an alternative to CYC. In a prospective study on the RAVE trial’s patients, using RTX showed promising results with a safe and more effective outcome in relapsed patients who were treated with either RTX or CYC [40]. In another comparative study of RTX and CYC on 16 GPA and 20 MPA patients, RTX was favorable for the induction of remission in the refractory AAV patients [96]. RTX is considered a promising treatment for elderly patients due to the higher risk of treatment-related side effects in AAV patients under CYC or GC treatment [83]. Additionally, RTX could provide an opportunity for the treatment of patients with high-grade CYC-related adverse events and be chosen as a first-line treatment [97]. Another study on 251 AAV patients with severe renal disease showed that although RTX and CYC both induce remission in AAV patients, CR was better induced by RTX than CYC [98]. A study on Hispanic patients showed that RTX and CYC have a similar effect on the induction of remission; however, RTX proved more effective in the prevention of relapse [82]. Although RTX was found to be as effective as CYC, it was found that RTX is more efficient than Infliximab, an anti-TNF-α agent. The analysis revealed that RTX provides better response and long-lasting CR was more achievable when compared to Infliximab. Moreover, upon the failed induction of remission in patients treated with Infliximab, RTX can induce remission in most patients [99].

### 3.6. Maintenance Regimen

Although RTX administration in active patients can induce disease remission, due to the repopulation of CD20+ B cells, most of the patients can show disease relapse during the first year. To avoid disease relapse in RTX-treated patients, preemptive maintenance therapy is being used. However, there is still controversy about whether patients benefit from maintenance therapy without showing any clinical symptoms of relapse and regarding efficacy, optimum dose, the interval between induction and maintenance infusions, and safety.

In a cohort study with 114 GPA patients, it was shown that administering preemptive maintenance RTX every 6 months for 18 months could lead to decreasing the daily dose of corticosteroids (median daily prednisone dose was 5 mg at 2 years in 76% of patients), a long time of disease remission (2-year relapse-free survival was reported in 85% within a median follow-up of 3.6 years) and relatively low rate of serious infection and serious adverse event rates (4.9 and 8.1 per 100 patient years, respectively) [37]. Regardless of RTX maintenance dose and intervals, a significantly higher relapse-free survival was previously reported in patients who received RTX maintenance as compared to those with a single course of RTX [100]. Frequently relapsed patients are hard-to-manage patients and it was shown that RTX therapy is effective and safe for chronic relapse patients for the induction of remission and maintaining the remission [101]. The effectiveness of maintenance therapy was also suggested by evaluating 795 GPA patients from the French Vasculitis Study Group Registry; it was shown that RTX maintenance therapy can remarkably help to achieve sustained remission (≥6 months) off therapy [102]. In another study including 66 GPA patients who received 500 mg RTX every six months until month 18, in a 34.2 ± 26.2-month follow-up, the relapse rate was 11.2/100 patient years. However, relapse occurred at a mean of 13.5 ± 14.7 months after the last RTX infusion [103]. This suggests a continuous need for RTX maintenance therapy.

Since disease relapse is because of the repopulation of B cells, patients can be either treated in a fixed-schedule manner (usually 6 months) or after the repopulation of B cells/detectable ANCA. To clarify this issue, the MAINRITSAN2 trial included 117 GPA and 45 MPA patients. Accordingly, as maintenance therapy, some patients received a 500 mg RTX infusion in the cases with the repopulation of CD19+B lymphocytes or those with the reappearance/rise of ANCA until month 18; controls received a fixed 500 mg dose of maintenance RTX every six months, regardless of cell population and ANCA titer. In both groups, at month 28, sustained remission was reported in most of the patients, while 14 of 81 (17.3%) in 13 conditional RTX-treated groups and 8 of 81 (9.9%) fixed-schedule patients showed disease relapse, which was not statistically significant. It is worth noting that those in the conditional RTX-treated group received less infusion than the control (248 vs. 381) [27]. The post hoc analysis of the MAINRITSAN2 trial was conducted to investigate the difference between the efficacies of the number of initial infusions of RTX. This analysis found no significant difference in the sustained remission rate within 12 months between patients administered RTX on the first and fourteenth days and those who only received a single infusion of 500 mg as an induction, which suggests the omission of the second infusion [104]. There are also some reports that show no significant difference between RTX maintenance therapy and conventional immunosuppressive agents. However, patients benefited from a reduced dose of corticosteroids [105].

In terms of the safety of preemptive RTX maintenance therapy, most of the studies did not find any serious issues. However, in a study performed by Besada et al. [106], despite the decreasing rate of relapse, because of side effects (mostly hypogammaglobulinemia and infections), RTX was discontinued in one-third of patients after a median of 41 months. However, during a median of 14.5 months, nobody showed disease relapse.

### 3.7. Combination Therapies

Following the study for the RITAZAREM trial, a high rate of disease remission in 188 relapsed AAV patients (90%) was reported after RTX therapy in combination with GC [107]. Geetha et al. [108] evaluated the possible additional benefit of adding CYC to the RTX+GC regimen in 37 AAV patients; they observed no significant differences in remission, and mortality by adding CYC to RTX and GC was observed.

Forty-nine patients with ANCA-positive state or renal-limited vasculitis were included in a prospective study. The patients were treated with low-dose CYC and RTX plus a 7-day or 14-day GC regimen. A short-term GC regimen or long-term GC regimen for the induction of remission showed a similar result; however, GC-associated adverse events were decreased [109]. Venhoff et al. [110] investigated 37 refractory AAV patients treated with RTX and conventional immunosuppressive therapies and showed that remission was achieved in almost all of the patients besides long-lasting B cell depletion. A retrospective French national study on 17 persistent AAV patients, who failed to respond to RTX or CYC, showed the high efficacy of RTX plus MTX combination therapy in persistently active GPA with predominant granulomatous manifestation [111]. One study explored 89 GPA patients and revealed that adding conventional agents (AZA, MTX, or mycophenolate mofetil) to RTX and GCs significantly decreased relapse rates without increasing adverse events [112]. Moreover, another study shows that combined RTX and CYC in refractory patients in severe relapse provides long-term remission and results in lowered CYC usage [113]. Adding CYC to the RTX developed more subsided immune agents that resulted in the rapid improvement of renal inflammation and function [114].

### 3.8. Organ Involvement

RTX was thought to be an effective induction treatment in small vessel vasculitis, mainly in GPA patients [115]. Complete remission was reached more frequently in GPA patients than in MPO [116]. Moreover, the PR3-ANCA patients in the RAVE trial were shown to experience sustained remission more than MPO patients [117]. Holle et al. [118] investigated the efficacy of RTX in 59 patients with refractory GPA and showed that 61.3% of patients achieved complete or partial remission. They also reported that progression was better achieved in vasculitis manifestation than in granulomatous symptoms.

#### 3.8.1. Ear, Nose, and Throat (ENT)

There are different studies on the efficacy of ENT involvement in AAV patients. In this regard, in a retrospective study, including 51 and 48 patients with and without a history of being treated with RTX, respectively, Lally et al. [119] showed that RTX is more effective than conventional therapies (i.e., AZA, MTX, CYC) in the induction of disease remission in patients with otolaryngologic manifestations. However, in another retrospective study containing 11 AAV patients, despite the significant decrease in the daily dose of prednisone, no significant improvement in patients’ otolaryngological complaints was reported [120]. In another retrospective study including 34 GPA patients with head and neck manifestations, 88% of patients responded partially or completely to the RTX within 6 months. The other 12 percent of patients achieved remission following the infusion of the second course of RTX.

#### 3.8.2. Ocular

Several studies have investigated the efficacy of RTX in granulomatous inflammation of the eye in patients with AAV. In this regard, Lower et al. [121] showed in nine patients with refractory granulomatous eye disease that RTX is an effective agent with corticosteroid-sparing properties. Baslund et al. [122] studied 10 patients with orbital inflammation and showed that RTX therapy ameliorates visual deficiency, granuloma volume, and other manifestations. Moreover, Pelegrin et al. [123] showed that RTX therapy was effective in patients with orbital inflammation who are resistant to conventional immunosuppressive therapies. Moreover, another study showed that 80% of patients with retro-orbital granulomas responded to RTX and RTX reduced exposure to immunosuppressive drugs [124]. A retrospective study on 59 AAV patients with orbital mass showed that RTX successfully treated more patients than CYC [125]. RTX is effective in refractory ophthalmic GPA patients [126]. In a retrospective study on 63 AAV patients with ocular involvement, scleritis and episcleritis were the prominent manifestations, and in about 14 percent of patients, ophthalmic disorders were the earliest manifestations of GPA. RTX therapy was shown to be more effective than CYC in refractory patients [127]. Another retrospective study on 37 patients with ocular manifestations of AAV showed that RTX was effective in the induction of remission within 36.5 months of follow-up [128]. Although RTX provided a good response in ocular AAV patients, it should be noted that the developing retroorbital fibrosis due to the lack of inflammation resulted in diminished visual accuracy [126].

#### 3.8.3. Renal

A retrospective study on 155 AAV patients with severe kidney disease showed that conventional therapies (CYC and GCs) responded to 51% of patients [129]. In a multicenter retrospective survey of 14 patients with positive or negative ANCA states, including those with severe renal disease, it was shown that RTX and GCs without CYC induced high-rate remission in patients and dialysis was terminated in 71% of dialysis-required patients [130]. A retrospective study on 153 AAV patients with GN showed that no significant difference was detected between RTX and CYC for the induction of remission in AAV patients with severe GN. Although no remarkable difference was observed in the setting of the remission rate, the percentage of patients without dialysis requirements in the RTX group was more than in the CYC group. However, RTX was shown to be more effective and resulted in fewer adverse events than conventional therapies in AAV patients with renal involvement [131,132] and RTX could be considered as an alternative treatment in patients resistant to conventional therapies [133,134,135,136].

Chocova et al. [137] studied 18 patients with AAV and showed that following RTX therapy, approximately 72% of patients experienced remission (CR or PR), and patients with ENT, lung, or renal involvement were not in CR. In 2011, a study on 23 AAV patients with severe renal involvement investigated the long-term efficacy of low-dose CYC combined with two infusions of 1 g RTX with a 2-week interval regime. This study showed that low-dose CYC plus RTX provides prolonged, relapse-free remission and decreased glucocorticoid usage, which reduced adverse events [138]. Later, a prospective cohort study on 66 renal AAV patients within 56 months of follow-up showed that a combined RTX-CYC-GC regimen as induction therapy followed by AZA and tapered GCs showed better responses than the conventional strategy of care. A total of 94% of patients achieved disease remission in 6 months and disease modulation was achieved earlier, while 70% of patients remained relapse-free over 5 years [139]. Another study was conducted on eight MPO-AAV patients with severe renal disease treated with a CYC-free RTX regimen. In this study, RTX provided an incomplete renal function improvement. Lack of efficacy was shown in cases that were previous non-responders and RTX recovery function was not related to the reaction to CYC [140]. Roccatello et al. [132] studied 25 AAV patients and compared the efficacy of two regimens, including RTX+CYC+GC without maintenance therapy (RTX group) and CYC+GC with AZA maintenance therapy (conventional group). CR was achieved in 93% of the RTX group and 70% in the conventional group; however, no significant difference was observed between these two groups in 6-month and 12-month renal recovery. Another study on 36 AAV patients with renal involvement showed that RTX is a favorable induction agent in relapsed AAV patients, and 41.7% and 28.6% of AAV patients under CYC and RTX faced ESRD or death, respectively [96]. Moura et al. [98] investigated the efficacy of RTX and CYC in 251 patients with severe disease and showed that no significant difference was observed between RTX and CYC in the induction of remission. However, CR within 6 months was better achieved in patients treated with RTX than with CYC, and the rate of ESRD and dialysis requirements were significantly lower in RTX-treated patients. As mentioned in the literature, the optimal regimen was not proven and further studies are needed to elucidate the best strategy in AAV with renal involvement.

#### 3.8.4. Respiratory Tract

Tracheobronchial stenosis (TBS), including subglottic stenosis (SGS) and bronchial stenosis (BS), is one of the severe manifestations of Wegner. TBS responds poorly to GCs and conventional immunosuppressive therapies [141]. In a retrospective study on 26 patients with TBS, the remission rate of patients treated with RTX was 80%, while CYC provided remission in 20% or 17% of patients, with or without local interventions, respectively. CYC is effective in the treatment of BS, but RTX is recommended for SGS [142]. A pulmonary nodule is a common presentation of GPA; however, Henderson et al. [143] demonstrated that in GPA patients with pulmonary nodules who did not respond to conventional immunosuppression, RTX can remove the nodules and miniaturize the large nodules. A study on 19 Hispanic AAV patients with diffuse alveolar hemorrhage showed that a regimen of CYC as initial therapy followed by RTX and plasmapheresis is superior to CYC plus plasmapheresis without RTX and provides six months of survival [82].

**Table 2 biology-11-01767-t002:** Non-randomized studies of RTX therapy in AAV patients.

Author	Country	Type of Study	Number of Patients	Median Age	Sex	BVAS Score	Dose of RTX	Induction or Maintenance of RTX	Special Condition	CR or PR	Details
Thietart et al., 2022 [144]	France	Prospective	52 GPA 41 MPA	79.4	51F/42M	NG	4 × 375 mg/m^2^, 2 × 1 g (induction)/500 mg (maintenance)	Both	Elderly patients	57/66 remission	RTX therapy achieves and maintains remission in most elderly patients
Brogan et al., 2022 [84]	Multicenter	Non-randomized uncontrolled clinical trial	19 GPA, 6 MPA	17	20F/5M	* PVAS: 8	4 × 375 mg/m^2^	Induction	Pediatrics	All CR	RTX is effective and safe for children
Roccatello et al., 2022 [132]	Italy	Retrospective	15 case: 11 MPA, 4 GPA; 10 control: 8 MPA, 2 GPA	69 ± 11.6 (case), 72 ± 12.4 (control)	15M/10F	21 (case), 23 (control)	6 × 375 mg/m^2^	Induction	Renal involvement	14/15 CR 7/15 CR	No significant difference between RTX+CYC+GC and CYC+GC+AZA in renal recovery
Moollan et al., 2022 [96]	Ireland	Retrospective	20 MPA, 16 GPA	63.5	23M/13F	15	NG	Induction	Renal involvement	NG	RTX is favored as an induction agent for relapsed AAV compared to CYC
Loftis et al., 2022 [82]	US	Retrospective	12 MPA 7 GPA	55	6M/13F	25	NG	Induction	Hispanic patients	14 treated with RTX (10 remission, 4 died)	RTX is superior to CYC in preventing relapse
Besade et al., 2013 [106]	Norway	Retrospective	35 GPA	48	19M/16F	9	2 × 1 g	Both	Maintenance therapy with RTX	29 CR 4 PR	Long-term RTX was effective in prevention of relapse
Yusof et al., 2015 [90]	UK	Prospective	35 AAV	56	17M/18F	10.5	2 × 1 g	Induction	Relapsing AAV	20/35 CR 13/35 PR 2/35 NR	RTX provides long-term response
Baslund et al., 2012 [122]	Denmark	Prospective	10 GPA	50	3M/7F	NG	2 × 1 g	Induction	Orbital inflammation	10 remissions	RTX should be considered for dangerous ophthalmic patients
Wendt et al., 2011 [41]	Sweden	Retrospective	14 GPA 1 MPA 1 EGPA	60	9M/7F	9.5	4 × 375 mg/m^2^, 2 × 1 g, 2 × 0.5 g	Induction	Relapsing or refractory AAV	12/16 CR 3/16 PR 1/16 died	RTX provides long-term remission
Venhoff et al., 2014 [110]	Germany	Retrospective	32 GPA 5 MPA	62	21M/16F	13	2 × 1 g, 2 × 0.5 g	Induction	Relapsing or refractory AAV	19/37 CR 16/37 PR 1/37 NR 1/37 lost	RTX following conventional IST as maintenance therapy provided good response
Moog et al., 2014 [89]		Retrospective	15 GPA 2 MPA	58	10M/7F	13	1 × 375 mg/m^2^	Induction	Single-dose RTX therapy	6/17 CR 11/17 PR	Single-dose RTX plus other IS are less effective than lymphoma protocol
Wawrzycka-Adamczyk et al., 2014 [87]	Poland	Retrospective	12 GPA	50	5M/8F	9.5	Median 1 g	Induction	Refractory GPA	11/12 remission	Low-dose RTX is effective for refractory AAV
Chasseur et al., 2020 [145]	Belgium	Retrospective	48 GPA 9 MPA	57	29M/28F	6.2 ± 2.5	-	Induction	RTX with and without GC therapy	34 CR (GC and RTX) 0 CR (RTX)	Low CR rate using RTX without GC
Brihaye et al., 2007 [64]	France	Retrospective	8 GPA	49.6	5M/3F	14.3	4 × 375 mg/m^2^, 2 × 1 g	Induction	Relapsing/refractory GPA	3 CR 3 PR 2 NR	RTX plus steroid improved clinical outcome
Takeyama et al., 2021 [105]	Japan	Retrospective	107	73	44M/79F	13	375 mg/BSA	Both	Comparing RTX, other IST, and GC alone	1- and 2-year relapse-free was 92.9% and 84.4%	RTX maintenance therapy is effective and provides lower dose of GCs
Asín et al., 2019 [127]	France	Retrospective	63	46.1	33M/30F	NG	375 mg/BSA	Induction	Ocular manifestation	80.9% remission	RTX induced remission in refractory patients and was more effective than CYC
Caroti et al., 2019 [140]	Italy	Retrospective	8	54	1M/7F	14	4 × 375 mg/m^2^, 2 × 1 g	Both	Renal involvement	8/8 remission	RTX was effective in partial renal function recovery
Mansfield et al., 2011 [138]	UK	Prospective	13 GPA 10 MPA	59	12M/11F	21	2 × 1 g	Induction	Severe renal involvement	23/23 Remission	RTX+low-dose CYC is effective in the induction of remission
P.McAdoo et al., 2019 [139]	UK	Prospective	66	62	38M/28F	19	2 × 1 g	Induction	Renal involvement	94% remission	Combination of GC, RTX, CYC is better than previous regimen
Menthon et al., 2011 [99]	France	Prospective randomized	17 (8 RTX, 9 IFX)	52.9 ± 17	8M/9F	12.6	4 × 375 mg/m^2^	Both	Refractory WG	3 CR 1 PR	RTX provides a higher rate of response and longer-lasting CR than Infliximab
Stasi et al., 2006 [31]	Italy	Prospective	8 GPA 2 MPA	53	5M/5F	5.5	4 × 375 mg/m^2^	Induction	Relapsing or refractory	9 CR 1 PR	RTX was effective in severe patients or non-responders to standard treatment
Lally et al., 2014 [119]	US	Retrospective	99	49.8 ± 15.1	31M/68F	NA	-	Maintenance	ENT manifestation	No ENT active disease in 92.4%	RTX results in 11-times lower rate of active symptoms than conventional therapies
Del Pero et al., 2009 [124]	UK	Retrospective	34 GPA	47.1	21M/13F	11	4 × 375 mg/m^2^, 2 × 1 g	Induction	ENT manifestation	21/34 CR 9/34 PR 4/34 NR	RTX provides 80% response, reduced exposure to other therapies
Casal Moura et al., 2020 [98]	US	Retrospective	251 (64 RTX, 161 CYC)	66	128M/123F	8	4 × 375 mg/m^2^	Induction	Severe renal disease	54/64 remission	Risks and benefits of CYC and RTX were balanced; adding PLEX provides no benefit
Girard et al., 2015 [142]	France	Retrospective	26	32	9M/17F	8	NG	Induction	Tracheobronchial stenosis	80% remission in RTX group, 42% remission in CYC group	RTX was more successful in achieving remission than CYC
Miloslavsky et al., 2014 [40]	US	Retrospective	24 GPA 2 MPA	NG	NG	5.3	4 × 375 mg/m^2^	Induction	Re-treatment of AAV relapse	17/26 CR 8/26 PR 1/26 died	Re-treatment of relapsed patients with RTX/GC was effective
Lionaki et al., 2017 [113]	Greece	Retrospective	29 GPA 6 MPA	48.6	15M/20F	15.1	4 × 375 mg/m^2^, 2 × 1 g	Induction	Refractory relapsing AAV	21.4% CR 78.6% PR	CYC plus RTX provides prolonged remission and less CYC usage
Cartin-Ceba et al., 2012 [101]	US	Retrospective	53 GPA	46	25M/28F	5	4 × 375 mg/m^2^, 2 × 1 g	Both	Chronic relapsing GPA	52/53 CR 1/53 PR	RTX was effective for induction and maintenance
Calich et al., 2014 [103]	France	Retrospective	66 GPA	50 ± 17.4	32M/34F	9.5	4 × 375 mg/m^2^, 2 × 1 g, 500 mg every 6 m	Both	Low-dose RTX as maintenance therapy	25/66 CR 27/66 PR 14/66 NR	Low-dose RTX provides low-level rate of relapse
Lovric et al., 2009 [35]	Germany	Retrospective	13 GPA 1 MPA 1 EGPA	45	8M/7F	12	4 × 375 mg/m^2^	Induction	Refractory or relapsing AAV	6/15 CR 8/15 PR 1/15 NR	RTX was effective to induce remission
Mittal et al., 2021 [43]	India	Retrospective	77 GPA	40	28M/49F	12	2 × 1 g, 4 × 375 mg/m^2^, 0.5 g every 6 m	Both	RTX as induction and maintenance	60% remission	RTX was effective as an induction and maintenance agent
Smith et al., 2012 [44]	UK	Retrospective	61 GPA 12 MPA	52	30M/43F	DEI = 4	2 × 1 g, 4 × 375 mg/m^2^, 0.5 g every 6 m	Both	Refractory or relapsing AAV	61/73 CR 8/73 PR 4/73 NR	Fixed-interval RTX therapy reduces relapse risk
Ramos-Casals et al., 2010 [33]	Spain	Prospective	17 GPA 2 MPA	46.2	9M/10F	NG	-	-	Severe refractory patients	10/19 CR 3/19 PR 6/19 NR	RTX can be used for severe refractory patients
Rees et al., 2011 [34]	UK	Retrospective	15 (11 GPA)	47.2	9M/6F	Median BVAS 13	4 × 375 mg/m^2^, 2 × 1 g	Induction	Refractory vasculitis	All CR	RTX was effective as an induction agent
Holle et al., 2012 [118]	Germany	Retrospective	59	54	35M/24F	16	4 × 375 mg/m^2^	Induction	Comparing efficacy of RTX in granulomatous vs. vasculitis symptoms	61.3% CR	RTX was more effective in vasculitis than granulomatous manifestations
Gulati et al., 2021 [114]	UK	Retrospective cohort	64	66	39M/25F	19	2 × 1 g	Induction	Combined therapy of severe AAV patients	94% CR	Combined RTX, low-dose CYC, and GC resulted in early and sustained remission
Pepper et al., 2019 [109]	UK, Ireland	Prospective	49	65.5	24M/25F	16.4	2 × 1 g	Induction	GC-free regimen in severe AAV patients	45/46 remission	Rapid GC discontinuation in severe AAV is effective as standard therapy with fewer adverse events
Puéchal et al., 2021 [102]	France	Retrospective	434	53.4	252M/182F	15.3	NG	Both	10 years follow-up to assess prolonged remission without treatment	-	Continual remission was significantly more achieved by RTX than conventional therapies
Joshi et al., 2015 [128]	UK	Retrospective cohort	37	51.5	-	-	2 × 1 g	Induction	Ocular GPA	31/37 CR 5/37 PR	RTX effectively induces remission of ocular manifestation in AAV patients
Malm et al., 2014 [120]	US	Retrospective	11	30	5M/6F		4 × 375 mg/m^2^	Induction	Otolaryngologic manifestation	No improvement in otolaryngological manifestations	RTX did not provide improvement in ENT manifestations
Kant et al., 2019 [85]	US	Retrospective	9	63	3M/6F	15	4 × 375 mg/m^2^	Induction	AAV patients with GN	All remission	Consecutive therapy with GC and CYC followed by RTX was effective
Shah et al., 2015 [130]	US, Sweden, UK	Retrospective	6 GPA 8 MPA	61	8M/6F	NG	4 × 375 mg/m^2^, 2 × 1 g	Induction	Severe renal disease	14/14 responded	RTX plus GC regimen induced high-rate remission and dialysis withdrawal
Sorin et al., 2022 [111]	France	Retrospective	17 GPA	-	-	-	-	Induction	Persistent active GPA with granulomatous manifestations	88% remission	Combined RTX/MTX induced remission
Lower et al., 2012 [121]	US	Retrospective	5 AAV 4 sarcoidosis	54.3	2M/3F	NG	375 mg/m^2^, 2 × 1 g	Both	Refractory ocular patients	5/5 responded	RTX was effective in the treatment of patients with eye disease.
Taylor et al., 2009 [126]	UK	Retrospective	10 GPA	48.2	5M/5F	NG	2 × 1 g	Induction	Refractory ocular patients	10/10 Remission	prolonged remission was achieved using RTX
Pullerits et al., 2012 [133]	Sweden	Retrospective	28 GPA 1 MPA	49.3	15M/14F	6	4 × 375 mg/m^2^	Induction	Refractory AAV	21% CR 41% PR 38% NR	RTX is an alternative option in conventional treatment-resistant patients
Chocova et al., 2015 [137]	Czech Republic	Prospective	15 GPA 3 MPA	37.5	11M/7F	9.5	4 × 375 mg/m^2^, 2 × 1 g	Induction	Relapsing or refractory AAV	8/18 CR 5/18 PR 2/18 NR 2/18 died 1/18 lost	RTX is associated with remission in 72% of patients
Keogh et al., 2006 [30]	US	Prospective	10 GPA	57	7M/3F	6	4 × 375 mg/m^2^	Induction	Severe refractory GPA	10 CR	RTX was an effective induction agent for severe refractory GPA
Mc Gregor et al., 2015 [91]	US	Prospective	56 GPA 52 MPA 3 EGPA 9 renal	50	55M/64F	NG	4 × 375 mg/m^2^, 2 × 1 g	Induction	-	103 remissions	Two-dose and >2-dose RTX regimens were similar
Roccatello et al., 2011 [136]	Italy	Prospective	5 GPA 4 MPA 2 EGPA	57.5	6M/5F	22	6 × 375 mg/m^2^	Induction	Refractory patients	11 remissions	RTX was effective in patients resistant to conventional IST
Roccatello et al., 2017 [93]	Italy	Prospective	5 GPA 4 MPA 2 EGPA	57.5	6M/5F	22	6 × 375 mg/m^2^	Induction	Long-term follow-up of refractory patients	4/11 remained in remission 7/11 re-treated	RTX in 6 doses of 375 mg/m^2^ provides long-term remission
Ayan et al., 2018 [135]	Turkey	Retrospective	21 GPA 4 idiopathic	44	11M/14F	NG	4 × 375 mg/m^2^, 2 × 1 g	Both	Untreated patients with conventional IST	18 CR 1 died 6 ongoing diseases	RTX was effective in patients resistant to conventional IST
Knight et al., 2014 [39]	Sweden	Retrospective	12 GPA	52	5M/7F	9	2 × 1 g	Maintenance	Severe relapsing AAV	11/12 remission 1/12 NR	RTX therapy every six months is an effective maintenance treatment
B.Jones et al., 2009 [86]	UK	Retrospective	46 GPA 10 MPA 5 EGPA 4 other	47	34M/31F	DEI 4	4 × 375 mg/m^2^, 2 × 1 g	Induction	Comparing different regimes of RTX	49/65 CR 15/65 PR 1/65 NR	No difference between the two regimes
Brihaye et al., 2007 [64]	France	Retrospective	8 GPA	49.6	5M/3F	14.3	4 × 375 mg/m^2^, 2 × 1 g	Induction	Relapsing/refractory GPA	3 CR 3 PR 2 NR	RTX plus steroids improved clinical outcome
Durel et al., 2019 [125]	France	Retrospective	56 GPA 1 MPA 2 EGPA	46	26M/33F	9	NG	Induction	Orbital mass	64% remission with RTX vs. 26% with CYC	RTX was more effective than CYC
Timlin et al., 2015 [83]	US	Retrospective	19 GPA 12 MPA	71 ± 6	10M/21F	4.4	4 × 375 mg/m^2^, 2 × 1 g	Induction	AAV patients older than 60	30/31 remission 1/31 NR	Elderly patients responded effectively to RTX
Puéchal X et al., 2019 [37]	France	Retrospective	114 GPA	52	40M/64F	9	500 mg every 6 m	Maintenance	Low-dose RTX as maintenance therapy	86% remission	Sustained remission using RTX for induction and low-dose maintenance
Azar et al., 2014 [112]	US	Retrospective	105 GPA	49	50M/55F	4	4 × 375 mg/m^2^, 2 × 1 g	Induction	Evaluation of RTX with or without other maintenance therapies	95/100 CR 1/100 PR 2/100 NR 1 died 1 lost	Conventional therapies plus RTX decrease relapse rate without increasing adverse events
Charles et al., 2013 [100]	France	Retrospective	70 GPA 7 MPA 2 Renal restricted 1 EGPA	54 ± 17	NG	7	4 × 375 mg/m^2^, 2 × 1 g	Both	Long-term follow-up	66% CR 25% PR	RTX was more effective as a maintenance therapy
Roll et al., 2012 [38]	Germany	Retrospective	50 GPA 8 MPA	50.2	28M/30F	NG	4 × 375 mg/m^2^, 2 × 1 g	Induction	Refractory AAV	22/58 CR 29/58 PR 4/58 NR	RTX was effective in refractory AAV

* Pediatric Vasculitis Activity Score; IST: immunosuppressive therapies; RTX: rituximab; CYC: cyclophosphamide; GPA: granulomatous with polyangiitis; MPA: microscopic polyangiitis; EGPA: eosinophilic granulomatous with polyangiitis; CR: complete remission; PR: partial remission; NR: not remission; NG: not given; NA: not assessed; F: female; M: male.

### 3.9. Resolving Rare Clinical Manifestations of GPA by Rituximab

Following the first report of the successful treatment of AAV with RTX, different rare clinical manifestations were reported as having been successfully treated with RTX. Some of the most important rare clinical presentations in GPA patients, which resolved following RTX, include retrobulbar granuloma [78], refractory ophthalmic [126], pachymeningitis [146], the presence of orbital inflammation [122], long-established end-stage renal disease [147], post-kidney-transplantation glomerulonephritis-related AAV [148], pituitary GPA [149], renal mass [150], gastric ulcer [151], pancreatitis [152], ophthalmoplegia [153], pyoderma gangrenosum [154,155,156], prostatitis [157] aortic inflammation [158], intestinal involvement [159], mastitis [160], aortitis [161], gingivitis [162], Isolated Pauci-Immune Pulmonary Capillaritis [163], isolated orbital GPA [164], cranial neuropathy [165], severe necrotizing scleritis [166], genital necrosis and inflammation [167], GPA-associated subcutaneous cheek odule [168], pseudo-tumoral digital nodules [169], hypertrophic pachymeningitis [170,171], acute myocarditis [172], palpable purpura [173], central nervous system vasculitis [174], intra-cranial hypertension [175,176], tracheobronchial stenosis [142], acute aortic valve regurgitation [177], GPA-mimicking meningeal tuberculosis [178], gynecological involvement [179], GPA-mimicking lung malignancies [180], CNS ischemic lesions [181], leukocytoclastic vasculitis-induced penile necrosis [182], stem cell transplantation [183], progressive skull base osteomyelitis [184], severe bilateral sensorineural hearing loss [185], nasal septal abscesses [186], oral and skin ulcer [187], nasal septal abscesses [186], membranous nephropathy [188], ventricular tachycardia [189], and hydralazine-induced AAV [190].

### 3.10. Unsuccessful Reports in Rituximab Therapy for GPA Patients

Despite several reports implying the high efficacy of RTX in controlling GPA, the first report of RTX therapy with a lack of efficacy in such patients was published by Aries et al. in 2006. From eight included patients, the disease activity remained unchanged in three patients and two of them even experienced disease progression, even when their peripheral blood B cells fell to zero [70]. In another study, two of eight patients did not respond to RTX and showed that vasculitis manifestation responded differently from granulomatous symptoms [64]. Moreover, Levric et al. [35] reported that one of the 15 patients with granulomatous AAV did not respond to RTX. In a study, RTX was not found effective in the improvement of patients’ otolaryngological complaints [120]. Caroti et al. [140] studied AAV patients with severe renal disease and showed that three of eight patients did not respond to the RTX. Lovric et al. [35], in their study, showed that one in 15 patients with upper-airway and eye involvement did not respond to RTX. In another study on AAV patients with ENT manifestation, 12% of patients did not respond to RTX; however, these patients achieved remission following the infusion of the second course of RTX [124]. However, another study on 11 patients with otolaryngologic manifestations did not respond to RTX and no improvement was observed in their manifestations within 23.5 months of follow-up [120]. In a multicenter study in the UK on 65 patients with refractory AAV, 98 percent achieved remission, but one patient showed no response to RTX [86]. Alesaeidi et al. [191] reported that a patient experienced pancreatitis and non-cancerous pancreatic mass as a non-first-line presentation of Wegner. Although RTX therapy resolved the complications, serious lethal infections appeared and patients were declared dead one month after the termination of RTX therapy. Paradoxical reactions to RTX were reported in some autoimmune diseases such as pemphigus vulgaris [192,193]. Regarding AAV, a study reported that six AAV patients experienced worsening disease following RTX treatment; however, they suggested not preventing the use of RTX in patients due to early complications [194]. Moreover, another study demonstrated that patients encountered exacerbation of ocular manifestation of AAV after injections of RTX [195,196], which may be due to rapid B-cell depletion causing cytokine release and the exacerbation of inflammation [197,198,199]. It is suggested that RTX should be injected with caution, especially in patients who have optic neuritis or lesions near the optic nerve. An AAV patient with upper-airway and eye disease was reported to show no response to RTX [35]. In patients with active refractory GPA, B-cell depletion by RTX is not related to the reduction in ANCA titer or the significant recovery of clinical features [70]. Collectively, based on the granulomatous or vasculitis manifestations, RTX was shown to be more effective in patients with vasculitis rather than patients with granulomatous manifestations [118], which be attributed to the late response of RTX in localized GPA manifestations [200]. It is speculated that different inflammatory ambiance in granulomatous lesions is associated with long-term granuloma and fibrous tissue formation may result in resistance to RTX therapy [201]. Furthermore, fibrous tissue may prevent the well-dissemination of a drug [202] and be responsible for a lower rate of remission in granulomatous manifestations than vasculitis manifestations. In this regard, the debulking operation of masses combined with immunosuppressive therapies resulted in successful treatment [203]. Moreover, although RTX was effective in the depletion of B-cells in peripheral blood, in situ B-cells are resistant to RTX, which can be attributed to the in situ B-cell activating factor (BAFF) and survival of B cells [204].

### 3.11. Safety

Similar to other medications, patients may experience some drug-related side effects along with the improvement of symptoms. However, as was reported by Jones et al. [86], no clear influence of RTX on the frequency of serious adverse events suggests benefits for patients treated with RTX. Safety concerns related to RTX’s administration could be categorized as a time of occurrence in three major categories, including infusion reactions or very early-appearing side effects (before B cell depletion), subsequent adverse events following the partial or complete depletion of B cells, and long-term RTX-related side effects. In other words, RTX-related side effects can be categorized based on the type of complications in five major categories, including infections, cytopenia, hypogammaglobulinemia, malignancy, and hypersensitivity, and among them, infection is the most common side effect [124,135,205], which is divided into bacterial, viral, and fungal types [206,207]. The safety profile of the short-term and long-term follow-up of RTX therapy in AAV patients was reported similarly [208]. Moreover, the safety profile of RTX therapy in AAV was also similar to other autoimmune diseases [209] and no significant difference between RTX and CYC was observed in the total rate of adverse events [210]. In this section, we assorted the findings based on the type of complications to provide data for clinicians at the practice level for RTX therapy in AAV patients.

#### 3.11.1. Hypersensitivity

Regarding the first category, hypersensitivity to RTX is a very early reaction that could occur in approximately one-third of AAV patients [101]. The symptoms of hypersensitivity could be very severe [138] and the development of an urticarial rash [64], anaphylaxis from RTX [37], and swelling of the hands after 48 h [124] of rituximab infusion have been reported. Some of the hypersensitivity reactions were self-limiting reaction infusions and were not repeated [124].

#### 3.11.2. Hypogammaglobulinemia

Hypogammaglobulinemia has been another common complication of treatment with RTX [211]. This phenomenon was reported during the maintenance of RTX therapy in different studies [212]. It was reported that hypogammaglobulinemia was more frequent within the first six months of RTX therapy [213]. The incidence of hypogammaglobulinemia and its severe form was reported to be about 50% [214] and 5% of patients, respectively [115,213,215]. However, the incidence of hypogammaglobulinemia varied according to the indication for RTX, whereas it was observed in 46% of pediatric lupus, 71% of autoimmune CNS disease, 60% of ANCA-associate vasculitis patients, and 12% in miscellaneous groups [213]. Using conventional immunosuppressive therapies, long-term glucocorticoid usage, kidney involvement, female gender [216], and 1 g every six months compared to the 2 g every 12 months [217] increases the risk of hypogammaglobulinemia following rituximab therapy, while subglottic involvement and orbital involvement reduces the risk [217]. Tariq et al. [218] reported that 42% of AAV patients had IgG-hypogammaglobulinemia related to RTX therapy. It is noteworthy that low IgG levels at baseline play a major role in the development of hypogammaglobinemia [212,219]. Some studies have suggested that hypogammaglobinemia increased the risk of infection in many cases [45,215,220], whereas some studies have suggested that there is no association between hypogammaglobulinemia and infection, even in severe hypogammaglobulinemia [214,216,221]. It should be noted that low levels of different types of immunoglobulins have a different result on infections that require hospitalization, wherein a low level of IgG and IgA result in an infection that requires hospitalization and a low level of IgM does not [222]. Altogether, it seems that hypogammaglobulinemia increases the risk of infection, especially those requiring hospitalization [222] in the severe form of hypogammaglobulinemia.

#### 3.11.3. Infection

Infection is the most common complication reported following RTX therapy in AAV patients [135] and can be severe in 10% [212] or more [205] of patients. It is notable that AAV patients with concomitant infection are difficult to manage due to the increased risk of infection following immunosuppressive therapy with CYC and GCs. However, Gregersen et al. [223] showed that RTX is an alternative for AAV patients with a severe infection that is associated with no further severe infection alongside the remission of the patient.

RTX is applied as an induction of remission and maintenance treatment option. Thietart et al. [144] investigated the difference between serious infection related to the induction regimen and maintenance therapy and showed that the rate of serious infection is significantly higher in the induction of remission by RTX than in maintenance therapy. The interval between taking RTX and causing infection varied in different studies and patients, ranging from three months to one year after treatment [113,224,225], whereas severe infection mainly emerged one year after RTX therapy [224]. The incidence of infection was similar following RTX, GC, and other IS [105]. In addition, the infection rate was higher in patients treated with RTX plus CYC than in RTX alone [113]. The infection rate of AAV patients following RTX therapy was higher than RA, which is related to the systemic nature of AAV, damage to the airways that predisposed the airway to infection, and long-term specific IS drugs used for AAV [226]. It seems that some risk factors exist that increase the risk of infection, including old age [115,224], low body mass index [214], bronchiectasis, endobronchial involvement [227], kidney dysfunction [214,225], and nephritic disease treated with GC-RTX [214], which is a risks factor for other side effects as well as infection [115]. It was shown that using trimethoprim-sulfamethoxazole prophylaxis reduced the risk of infection [227].

The emerging infections following RTX therapy can be divided into bacterial, viral, and fungal infections. In addition, to increase the risk of de novo life-threatening infections, such as sepsis [32,41,228] and viral hepatitis [32], several studies warn about the risk of the reactivation of chronic infections. The most common site of infection is the respiratory tract, especially the lower respiratory tract, and also pneumonia was the most common side effect [43,108,118,229]; however, the involvement of the ear and nose was probable [40].

The most-reported infections are RTX-induced pneumonia (pneumocystis pneumonia (PCP)) [230] after six months and pneumocystis jiroveci pneumonia (PJP) after 3 months [231], which can also be accompanied by bronchial stenosis [35] and other major chest infections such as tuberculosis [232], candida pneumonia, and bacterial pneumonia infections [83,124], which could be fatal. Atypical mycobacterial, salmonella [41], and urinary tract infections [32] have also been reported. The incidence of PCP is high [101], whereas one study found that a WG child developed PCP after 6 months of RTX therapy [230]. Therefore, it was recommended that PCP prophylaxis should be performed at least during the depletion of B cells [101,231]; although, the incidence of PJP is 1.2% and prophylaxis is not recommended [231].

Microorganisms that cause viral infections include CMV [32,206], JC virus, herpes zoster [108], varicella zoster, HBV [32,41,232] (even in HBsAg-negative patients), and HCV [35,43]. Elevated liver enzymes after 2 weeks [41] to 11 months [35] of rituximab therapy were observed; however, hepatitis can first appear in patients who had no cirrhosis and liver failure [43]. CMV-associated gastric ulcer [233] and retinitis [41], viral gastroenteritis [40], and JC virus-associated pneumonia [232] were observed in some patients after rituximab therapy. In addition, rituximab therapy can exacerbate the manifestation and reactivation of the JC virus, occurring within 3 to 40 months after treatment [222]. Treatment with rituximab also activated herpes zoster and resulted in pneumonia-associated herpes zoster virus [34,43]. The activation of varicella zoster occurred in five events after eight months of treatment [112]. Rituximab can increase the risk of fungal infections [207]. Lower respiratory tract infection was reported as the most common infection following RTX therapy and Invasive Aspergillus accounted for a quarter of lower respiratory infections developed after 8 months of rituximab therapy [112]. There are fewer common infections after rituximab therapy reported, including cutaneous abscess, otitis media, legionella pneumonia [210], fatal H1N1 flu infection [45], tuberculosis [232], urinary tract infections [32], gastrointestinal infection [214], fatal sepsis [41], vulvovaginal pyoderma gangrenosome [234], and atypical mycobacterial infection [32]. Accordingly, to prevent such complications or minimize them, it is recommended to consider screening for chronic infection [235] before RTX administration and the employment of preemptive therapies, and, if necessary, vaccinating against bacterial and viral pathogens, monitoring white cell count and immunoglobulin levels, and prophylaxis against more frequently reported infections [232].

#### 3.11.4. Cancer

Although there is no direct evidence for a carcinogenic effect of RTX, there are some sparse reports implying the possible carcinogenesis of RTX [32,124,138]. It was investigated that the risk of cancer in AAV patients was significantly higher than in the general population. It was shown that alongside a different pattern of cancer development based on age, sex, drugs, and ANCA type, the risk of lung and hematological cancer is mostly increased in AAV patients [236]. It is well-known that treating AAV patients with CYC increases the risk of malignancies [237], but it is clear that RTX is less carcinogenesis than CYC [238]. Rituximab therapy is one of the factors that increase the risk of cancer. It is reported that following RTX therapy, some different cancers developed, occurring several months or years after rituximab therapy, including squamous-cell carcinoma (SCC) of the esophagus or tongue [124], breast cancer [211], non-melanoma or melanoma skin cancers [90,112], bladder cancers, papillary thyroid cancer, uterine cancer, colon cancers [19], hematologic cancers, prostate cancer [32,91], lung cancer, peritoneal cancer, renal cell carcinomas, basal cell carcinoma, and hepatocellular carcinoma [91].

#### 3.11.5. Cytopenia

Cytopenia is another side effect of rituximab therapy, including leucopenia such as B-cell lymphopenia [20,35], thrombocytopenia [239], and neutropenia (late-onset) [240,241], which seem to be associated with B-lymphocyte depletion [241]. In some patients, neutropenia was observed shortly after the first dose [34,91] or the last dose [241], with some requiring hospitalization and pre-emptive antibiotic treatment [211] to control fever, but they did not need to receive granulocytes, whereas in one study it was reported that people with late-onset neutropenia (LON) improved with recombinant granulocyte colony-stimulating factor (GM-CSF) [42]. One study found that 11.9% of patients with GPA and MPA developed LON after an average of 86 days of rituximab therapy. Some of these patients developed neutropenia after the first dose of treatment and some after repeated courses of rituximab [240]. Leukopenia was observed in some patients, and some of them had transient leukopenia [21,35]. In the case of thrombocytopenia, a study reported that thrombocytopenia was observed in some patients after three weekly infusions of rituximab [83].

#### 3.11.6. Other Rare Side Effects

There are some less common side effects of rituximab, including acute myocardial infarction (AMI) [90], congestive heart failure (CHF) [88], ruptured aneurysm [108], venous thromboembolism (VTE) [210], anorexia nervosa [128], primary leukoencephalopathy (PML) [242], complications of orthopedic conditions such as hip replacement and bone fracture [211], pyomiosis [43], pyoderma gangrenosum [243], inflammatory vaginitis [244], herpes simplex stomatitis [222], Crohn’s disease [245], drug-induced pneumonitis [229], posterior reversible encephalopathy syndrome (PRES) [246,247], and transient visual disturbance [32]. Accordingly, to prevent such complications or at least minimize them, it was recommended to consider screening for chronic infection [235] before RTX administration and the employment of preemptive therapies; if necessary, vaccinating against bacterial and viral pathogens, monitoring white cell count and immunoglobulin levels, and prophylaxis against more frequently reported infections [232]. Alongside safety-related disorders, some reports of death were declared after RTX infusion, which can be mainly related to infection [210], cardiovascular complications [108], neurological complications [128], malignancy [248], respiratory failure [135], etc. [20,32,43]. It was also shown that the rate of death related to RTX therapy is higher when RTX is used as the induction of remission than maintenance therapy [144]. It should be noted that during RTX therapy, clinicians should consider these probable side effects and manage them carefully with regard to death occurring in the patients.

**Table 3 biology-11-01767-t003:** Safety profile of RTX therapy in AAV patients.

Side Effects	Comment	Ref.
Infection	PCP, PJP, TB, UTI, salmonella, atypical mycobacterial infection, influenza, legionella, cutaneous abscess, GI infection, vulvovaginal pyoderma gangrenosome. CMV, HBV, HCV, JC virus, HSV, herpes zoster, varicella zoster, aspergillus.	[32,34,35,40,41,43,45,83,101,105,108,112,113,115,118,124,135,144,205,207,210,214,223,224,225,226,227,228,229,230,231,232,233,234,235]
Hypogammaglobulinemia	Hypogammaglobulinemia and severe hypogammaglobulinemia were reported in about 50% and 5% of patients. Hypogammaglobulinemia-induced infection is a controversial issue. Baseline Ig level is a substantial factor in the development of hypogammaglobulinemia.	[45,115,119,211,212,213,214,215,216,217,218,219,220,221]
Cancer	Breast cancer, colon, hepatocellular, hematologic, uterine, thyroid, peritoneal, renal, bladder, lung, SCC of the tongue and esophagus, basal cell carcinoma, melanoma, and non-melanoma skin cancer.	[19,32,90,91,112,124,138,236,237,238]
Cytopenia	Leucopenia (B-cell lymphopenia), which can be transient; thrombocytopenia; neutropenia, which can be late-onset.	[20,21,35,42,83,211,239,240,241]
Hypersensitivity	Hypersensitivity reaction is a first-onset complication developed in one-third of injected patients. Hypersensitivity can emerge as different symptoms such as rash and swelling.	[37,64,101,124,138]
Other side effects	CHF, AMI, VTE, bone fracture, herpes simplex osteomatitis, visual disturbance, vaginitis, pyomiosis pyoderma gangrenosome, anorexia nervosa, PML, pneumonitis, Crohn’s disease, PRES, ruptured aneurysm.	[20,32,43,88,90,108,128,135,144,210,211,222,229,232,235,243,244,245,246,247,248]

## 4. Conclusions

Current randomized and non-randomized studies have shown that RTX was effective in AAV patients. AAV can manifest with different organ involvement and RTX was more effective regarding granulomatous manifestation. Furthermore, RTX can be considered an alternative treatment option when other conventional therapies are contraindicated. RTX is an effective treatment for the induction of remission and as maintenance therapy and successfully reduces the rate of relapse. RTX can also be associated with side effects including infection, cancer, hypersensitivity, cytopenia, and hypogammaglobulinemia, which should be noted when physicians prescribe RTX. In the clinical context, we suggest using RTX as a first-line treatment option, especially in patients with granulomatous manifestation, pediatrics, elderly patients, and those in whom other conventional immunosuppressive agents are contraindicated. More studies on the evaluation of combination therapies and a long-term follow-up of patients for monitoring safety could be useful to shed light on the other aspects of RTX.

## Figures and Tables

**Figure 1 biology-11-01767-f001:**
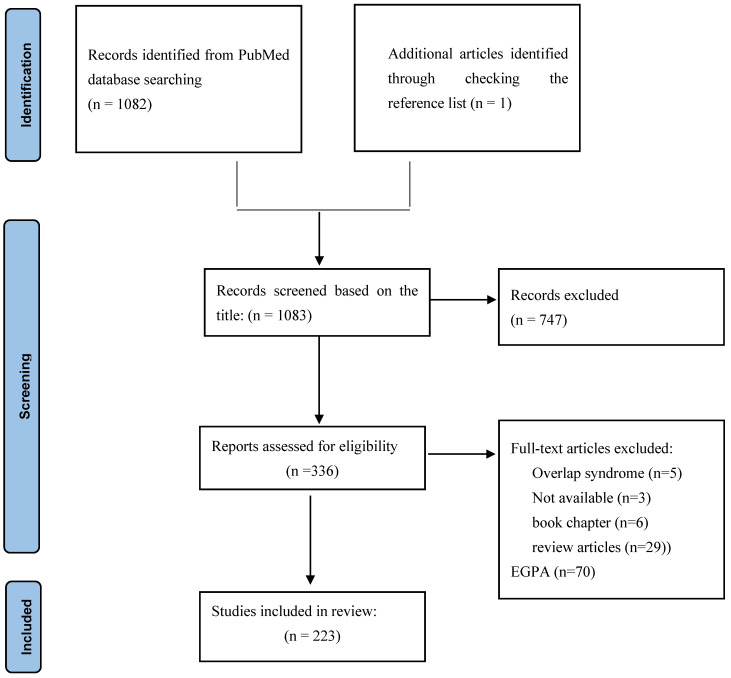
Preferred Reporting Items for Systematic Reviews and Meta-Analyses (PRISMA).

## Data Availability

The data are available within the article or are obtainable through a request to the corresponding author.

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
