# Peer review of "The Efficacy and Safety of Rituximab in ANCA-Associated Vasculitis: A Systematic Review"

_biology, 2022, doi:10.3390/biology11121767_

Round 1

Reviewer 1 Report

Dear authors

Thank you for this great work and very interesting paper

some minor comments:

Why not include GEPA? may be you can precise that in the method section

How was google scholar check done? can you precise that in the paper (keywords etc...)

Table1: could you give mean age and male/female ratio as in table 2

why include RATTRAP study in table 1 if data are not available? this did not add any value to your work

Maintenance regimen (3.6): Could you better precise if there is study  who showed that prolonged treatment with rituximab is indicated. Some patients are severe, relapsing...can we propose them a long treatment? infusion of rtx at 18 months, 24 months and even more? this may be interesting to discuss if there is data on that and if it is safe to perform it....

Reviewer 2 Report

In this manuscript, the authors methodically and thoroughly reviewed discoveries on rituximab (RTX), a monoclonal antibody depleting CD20+ B cells, as therapy for autoimmune disease ANCA-associated vasculitis (AAV) and concluded RTX as an effective and relatively safe therapeutic option for AAV. The topic would interest the readers of Biology studying autoimmune diseases. Only minor points need be addressed for acceptance:

1.      The chosen subtypes for the review, GPA or MPA need better introduction

2.      Result 3.1 about the strategy for paper collection might be redundant

Reviewer 3 Report

Manuscript submitted by Habibi et al aimed to investigate the efficacy and safety of rituximab (RTX) therapy in patients in patients with ANCA-associated vasculitis. And suggested that RTX can be considered an alternative treatment option when other conventional therapies are contraindicated. The systematic review is valuable for clinicians for medical experts who are focusing exploring the therapeutic strategy for AVV. Authors are asking to address the following questions for their paper to be considered for publication.

1. The abstract needs to be revised to improve the readability, especially for the abbreviations,

ANCA-associated vasculitis (AAV) is a rare multisystem involvement autoimmune disease due to the autoantibody production against human neutrophilic granulocytes (ANCA), including (PR3 and MPO)

Take above as an example, some of these abbreviations make quiet sense. And please use the full nomenclature throughout the text with the proper abbreviation when it first appears in the text.

2. Please include a section with all the full name and abbreviation to help reader to appreciate the work.

3. All data, for example, Line 222, should be presented as mean±stv to indicated the proper statistical analysis was conducted in the study.  

4. Please include the statistical analysis in the Method section.

5. For the sequence of each section, Section 3 is Results, Section 5 is Conclusion. Should Section 4 consider to be the Result or Discussion. Please check.
